

# Addressing incomplete lineage sorting and paralogy in the inference of uncertain salmonid phylogenetic relationships

Matthew A. Campbell[1], Thaddaeus J. Buser[2], Michael E. Alfaro[3] and J. Andrés López[1,4]

[1] University of Alaska Museum, University of Alaska—Fairbanks, Fairbanks, AK, USA
[2] Department of Fisheries and Wildlife, Oregon State University, Corvallis, OR, USA
[3] Department of Ecology and Evolutionary Biology, University of California, Los Angeles, Los Angeles, CA, USA
[4] College of Fisheries and Ocean Sciences, University of Alaska—Fairbanks, Fairbanks, AK, USA

## ABSTRACT

Recent and continued progress in the scale and sophistication of phylogenetic research has yielded substantial advances in knowledge of the tree of life; however, segments of that tree remain unresolved and continue to produce contradicting or unstable results. These poorly resolved relationships may be the product of methodological shortcomings or of an evolutionary history that did not generate the signal traits needed for its eventual reconstruction. Relationships within the euteleost fish family Salmonidae have proven challenging to resolve in molecular phylogenetics studies in part due to ancestral autopolyploidy contributing to conflicting gene trees. We examine a sequence capture dataset from salmonids and use alternative strategies to accommodate the effects of gene tree conflict based on aspects of salmonid genome history and the multispecies coalescent. We investigate in detail three uncertain relationships: (1) subfamily branching, (2) monophyly of *Coregonus* and (3) placement of *Parahucho*. Coregoninae and Thymallinae are resolved as sister taxa, although conflicting topologies are found across analytical strategies. We find inconsistent and generally low support for the monophyly of *Coregonus*, including in results of analyses with the most extensive dataset and complex model. The most consistent placement of *Parahucho* is as sister lineage of *Salmo*.

Corresponding author
Matthew A. Campbell,
macampbell2@alaska.edu

## INTRODUCTION

Within ray-finned fishes (Actinopterygii), there are numerous relationships that are challenging to resolve. Historically, anatomical characteristics have provided great insight into the evolutionary relationships of some groups, but failed to confidently resolve others (*Betancur-R et al., 2017*). In the past few decades, DNA sequence data has emerged as a method to gather numerous characters for phylogenetic inference. Sequence data generation in recent years has undergone dramatic and rapid evolution with respect to the cost and scale of DNA sequence determination and has fostered the production of

genomic-scale datasets for phylogenetic estimation. The theory and implementations of methods for analyzing these data have increased in complexity and performance broadly in parallel with increased data generation (*Muir et al., 2016*). The sheer quantity of data contained within genome-scale datasets has contributed to expectations that molecular phylogenetics may provide conclusive results regarding previously elusive evolutionary relationships. While additional anatomical investigation and the application of molecular phylogenetics has aided the resolution of some groups, such as the Scorpaenoidea (*Smith, Everman & Richardson, 2018*), counterexamples certainly remain. In particular, there are parts of the ray-finned fish tree of life that remain poorly resolved even with the application of high-throughput sequencing data sets of substantial numbers of loci. These persistently challenging-to-resolve relationships include those among early-branching euteleost lineages (*Campbell et al., 2017*) and the intra-clade relationships of Pelagiaria (*Campbell et al., 2018*; *Friedman et al., 2019*; *Miya et al., 2013*).

While conclusive resolution is desirable, evolutionary relationships can be expected to remain unresolved with molecular data under certain scenarios. For example, when the characteristics of the evolutionary process being investigated generated signals with ambiguous origins, prevented the formation and preservation of a phylogenetic signal, or when the methodological framework used to investigate that process is poorly suited for the data type. Discordance between gene trees or gene tree–species tree (GT–ST) conflict (Fig. 1) represents a special challenge in the inference of recalcitrant relationships because it may be the combined product of conflicting histories and functionally undetectable phylogenetic signal with difficult to disentangle contributions from each factor.

One predicted cause of GT–ST conflict is incomplete lineage sorting (ILS), which may be defined as the failure for two or more allelic lineages to coalesce within a population (*Degnan & Rosenberg, 2009*). Incompletely sorted diversity can lead to conflicting gene tree topologies if the time between speciation events is small and/or the population sizes are large (Fig. 2). The continued investigation of natural systems has emphasized the pervasiveness of hybridization (*Mallet, 2005*), and divergence times before or after hybridization may result in ILS and gene tree incongruence signatures identical to patterns exhibited by hybridization alone (*Yu et al., 2011*). In tree-based evolutionary analyses, hybridization events are very easily overlooked, and GT–ST conflict will be considered a consequence of ILS. Alternatively, in phylogenetic network analysis, such reticulations are largely taken as hybridization (*Yu et al., 2011*). Another potential source of GT–ST conflict is gene or genome duplication events. Genes that are duplicated, particularly those arising from whole-genome duplication (WGD) are frequently lost (*Berthelot et al., 2014*; *Scannell et al., 2007*). However duplication of a gene followed by loss of alternative copies in different lineages may lead to GT–ST conflict (Fig. 3). While massive gene loss is typical following WGD as polyploid organisms revert to a normal ploidy level (*Gerstein & Otto, 2009*), many genes in the genomes of lineages with ancestral genome duplications exist in a continuum from polyploidy to normal ploidy, a condition known as segmental polyploidy. In lineages where genomes exhibit segmental polyploidy, ILS can be expected
A. Species Relationships: ((I, II), III)

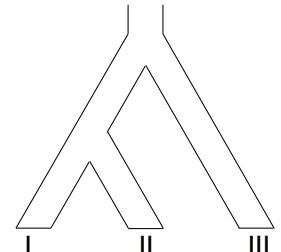

B. Possible Gene Trees

C. Different Gene Trees within Species Tree

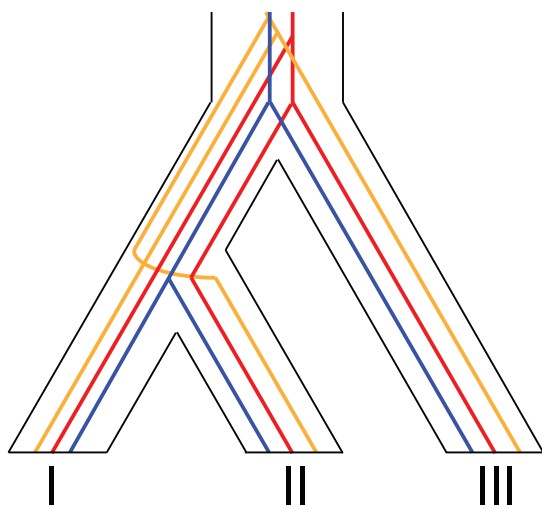

**Figure 1 Depiction of gene tree and species tree conflict.** (A) Hypothetical species relationships between three species, I, II and III. (B) Depiction of three possible gene trees. (C) Combined species tree and gene trees. Only one of the gene trees (blue) follows the species tree relationships.

to be more pronounced in the tetrasomic regions of genome compared to the disomic regions as their respective population sizes differ (Figs. 2B and 2D).

Whole-genome duplication events are known from across many eukaryotic lineages (*Campbell et al., 2016*) and some lineages have experienced successive rounds of WGD producing nested levels of ancestral polyploidy in the genomes of descendant lineages (*Meyers, Levin & Geber, 2006*; *Wendel, 2015*). Genomes of extant fishes are partly shaped by two rounds of WGD (commonly noted as 1R and 2R; Fig. S1) which are inferred to have taken place 450–600 million years ago (mya) (*Dehal & Boore, 2005*). A third well-supported WGD (3R) is inferred in the common ancestor of teleosts at around 350 mya (*Meyer & Van de Peer, 2005*). Additional WGD events are scattered through lineages in the fish tree of life (*Leggatt & Iwama, 2003*). A WGD occurred in the common ancestor of the Salmoniformes some time after the lineage diverged from the Esociformes. This WGD event occurred ~88 mya ago and it is known as the Ss4R (salmonid-specific fourth round) (*Berthelot et al., 2014*; *Macqueen & Johnston, 2014*).

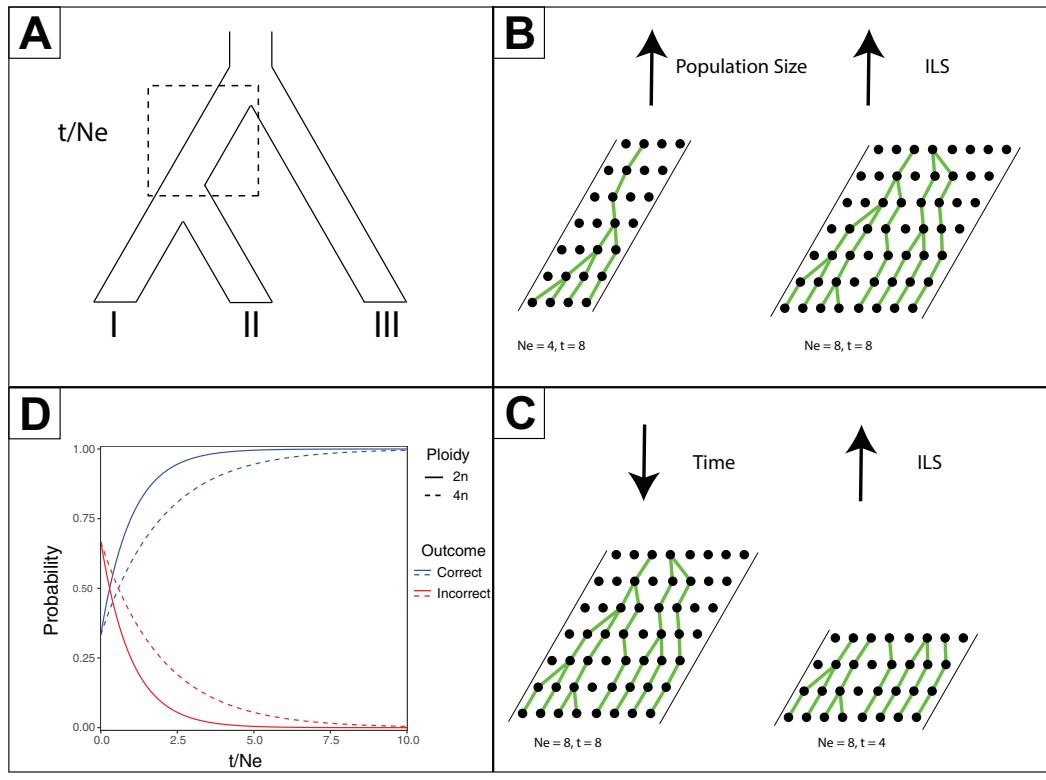

**Figure 2 Effects of time and effective population size ($N_E$) on incomplete lineage sorting (ILS).**
(A) Species tree depicting relationships between species I, II and III. The branch leading to the common ancestor of I and II is indicated in a box as where the ratio of time ($t$, in generations) versus $N_E$ is considered for the other subpanels. (B) Depiction of how increased population sizes increase ILS. A single coalescent event is depicted for each generation with population sizes of eight and four. (C) Depiction of decreased time increasing ILS. For a population of eight, four or eight generations are shown. (D) For a three-taxon species tree (as in Subpanel A), the probability of inferring a correct or incorrect gene tree is plotted as a function of $t/N_E$. As there are three possible outcomes, and only one correct one, the initial probability of inferring an incorrect gene tree is 2/3 at $t/N_E = 0$. The effect of different ploidy levels is shown, with increased ploidy increasing ILS.

Salmonids are a group of 228 fish species divided among three subfamilies: Coregoninae (whitefishes, ciscos, and innocu: 86 species), Thymallinae (graylings: 18 species), and Salmoninae (salmons, trouts, charrs, taimens: 124 species) (*Fricke, Eschmeyer & Fong, 2019*; *Nelson, 2006*). Despite the commercial and cultural importance of many salmonids, and intensive research on many of the commercially exploited species, questions about their evolutionary relationships remain unanswered. While monophyly of each of the three subfamilies is generally accepted, relationships between these subfamilies remain contentious. A sister-group relationship between Salmoninae and Thymallinae (S + T) has been supported by morphological (*Sanford, 1990*; *Wilson & Williams, 2010*) and molecular evidence (*Betancur-R et al., 2013*; *Shedko, Miroshnichenko & Nemkova, 2012*; *Yasuike et al., 2010*). In contrast, several molecular phylogenetic studies have offered support for a clade comprising Coregoninae and Salmoninae (C + S) (*Alexandrou et al., 2013*; *Crête-Lafrenière, Weir & Bernatchez, 2012*; *Near et al., 2012*). Finally, Coregoninae sister

A. Species Relationships: ((I, II), III)

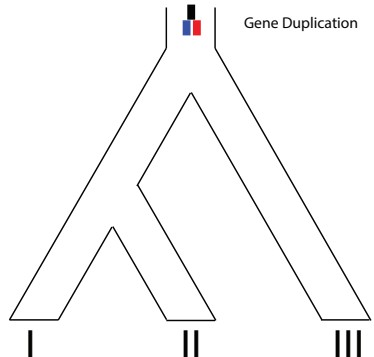

B. Duplication Followed by Loss

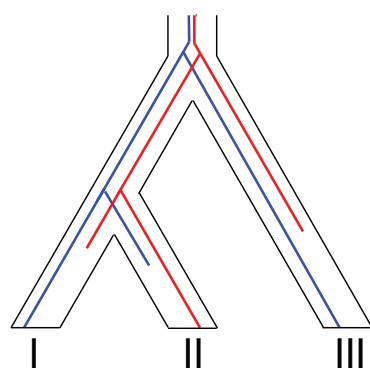

I & III would appear most closely related to each other

**Figure 3 Example of gene duplication and loss misleading phylogenetic inference.** (A) Species tree of three species depicting a gene duplication event in the common ancestor of all three species. (B) Example of loss of paralogs in different lineages. Each species exhibits a single copy of the formerly duplicated gene; however, construction of a phylogeny from those sequences would make it appear that species I and species III would be most closely related to each other.

to Thymallinae (C + T) has been indicated by several analyses (*Burridge et al., 2012*; *Campbell et al., 2013*; *Horreo, 2017*; *Li et al., 2010*; *Macqueen & Johnston, 2014*; *Robertson et al., 2017*). Thus, each of the three possible arrangements of the subfamilies has found some support in the literature. Another point of contention is the relationship between the coregonine genera *Coregonus* and *Stenodus*. Previous studies have variously placed *Stenodus* as sister to (*Bodaly et al., 1991*; *Horreo, 2017*; *Sajdak & Phillips, 1997*; *Vuorinen et al., 1998*) or nested within *Coregonus* (*Bernatchez, Colombani & Dodson, 1991*; *Crête-Lafrenière, Weir & Bernatchez, 2012*). A final point of debate is the placement of *Parahucho* within Salmoninae. Previous studies have variously placed *Parahucho* in a polytomy with *Oncorhynchus + Salmo + Salvelinus* (*Shedko, Miroshnichenko & Nemkova, 2012*) or as sister to: the genus *Salmo* (*Crespi & Fulton, 2004*; *Lecaudey et al., 2018*; *Matveev, Nishihara & Okada, 2007*; *Oakley & Phillips, 1999*); the genus *Salvelinus* (*Campbell et al., 2013*; *Crête-Lafrenière, Weir & Bernatchez, 2012*; *Horreo, 2017*); a clade composed of *Oncorhynchus + Salvelinus* (*Crête-Lafrenière, Weir & Bernatchez, 2012*); or a clade composed of *Salmo + Salvelinus + Oncorhynchus* (*Alexandrou et al., 2013*).

While phylogenetic uncertainty is not unique to salmonids, the numerous hypotheses of salmonid intrarelationships proposed in recent literature suggest serious challenges with molecular phylogenetic inference of the group and WGD may be a contributing factor (*Allendorf & Thorgaard, 1984*; *Berthelot et al., 2014*; *Lien et al., 2016*; *Macqueen & Johnston, 2014*; *Ohno, 1970*). Well-characterized genomes of salmonids reveal that 10–15% of their genomes are tetrasomically inherited, with high levels of overall genomic similarity throughout as a result of the Ss4R (*Lien et al., 2016*; *Pearse et al., 2019*). Within rainbow trout (*Oncorhynchus mykiss*), for example, these duplicated regions contain 2,278 tetrasomically inherited pairs of genes (*Campbell et al., 2019*; *Pearse et al., 2019*), meaning that 10.62% of the 42,884 annotated protein-coding genes in rainbow trout

are tetrasomically inherited. Comparative genome mapping shows broad conservation of partial tetrasomy across Salmoninae in eight chromosome arms, the "Magic Eight" with homologous regions in *Salmo*, *Salvelinus* and *Oncorhynchus* species indicated to be tetrasomic (*Brieuc et al., 2014*; *Campbell et al., 2019*; *Kodama et al., 2014*; *Lien et al., 2011*; *Sutherland et al., 2016*). Further analyses of recently produced genome assemblies and linkage mapping has continued to support the idea of the same genomic regions maintaining tetrasomy across salmonids (*Blumstein et al., 2019*). Genomic comparisons support that there are at least seven tetrasomically pairing chromosomes in salmonids, with possible intermediate pairs falling between tetrasomy and disomy present, and one tetrasomically pairing set identified only by linkage mapping. Tetrasomic recombination results in larger population sizes and duplicated loci for some genes and genome regions and may contribute to GT–ST conflict via ILS (Figs. 2B and 2D). While the path to rediploidization in salmonids has conserved patterns across species where rediploidization occurred prior to speciation, continued rediploidization within lineages after speciation has resulted in unique signatures—lineage-specific rediploidization or lineage-specific ohnologue resolution (LORe). These signatures are elevated in tetrasomically pairing regions of the salmonid genome (*Robertson et al., 2017*). Due to LORe, loci from tetrasomically pairing genomic regions of salmonid genomes are likely to have problematic orthology assignment. With a complex genome structure riddled with duplications and high-similarity regions that are undergoing independent rediploidization, the potential for the accidental inclusion of paralogous loci in phylogenetic analyses of this group is concerning (i.e., Fig. 3). Together both tetrasomy and lineage-specific rediploidization processes may be responsible for widespread GT–ST conflict in salmonids.

In this article we analyze a DNA sequence dataset from 500 Ultra Conserved Element (UCE) loci. These UCE loci were designed to be exchangeable across data sets and to be single copy across actinopterygiian fishes (*Faircloth et al., 2013*). One particular property of UCE loci that is, notable is that variation increases away from the core conserved region, providing variable sites across time scales and higher phylogenetic informativeness compared to protein-coding gene loci (*Gilbert et al., 2015*). The generous starting pool of 500 loci allows us to filter for paralogs or duplicates originating from the Ss4R while retaining relatively informative loci for phylogenetic inference. With UCE loci from representative salmonids, we test the following phylogenetic hypotheses while considering particularities of salmonid genome evolution: (1), the relationships of salmonid subfamilies, (2), the monophyly of the coregonine genus *Coregonus* and (3), the placement of the enigmatic salmonine genus *Parahucho*. To identify clear orthologs for analysis within salmonids from markers designed from diploid species, we filter loci based on assembled duplicates and on genomic location in known tetrasomic regions of salmonid genomes. We then test the effects of missing data and tetrasomic inheritance in phylogenetic analyses by generating alternative subsamples of the full dataset and compared the resulting inferred relationships with and without the multispecies coalescent.
## METHODS

### Sequence data acquisition and processing

We obtained sequence data from representatives of all major salmonid genera ($n = 15$), as well as from representatives of the two extant esociform families Esocidae and Umbridae, which are the closest diploid relatives of salmonids (*Ishiguro, Miya & Nishida, 2003*; *López, Chen & Ortí, 2004*). We generated genetic sequence data for phylogenetic analysis by sequence capture of 500 UCE loci following the protocol outlined for the 500 UCE actinopterygiian probe set (*Faircloth et al., 2013*) modified according to experimental conditions described in *Campbell et al. (2017)*. We sequenced these loci for 11 taxa and incorporated data for six additional taxa from previous studies described in Table S1 (*Campbell et al., 2017*; *Faircloth et al., 2013*).

Demultiplexed reads were trimmed of low-quality bases and filtered for minimum length with Trimmomatic version 0.32 using the following command line specifications and adapter fasta file available for download with the program: ILLUMINACLIP: TruSeq3-PE-splitAdapter.fa:2:30:10 LEADING:3 TRAILING:3 SLIDINGWINDOW:4:15 MINLEN:36 (*Bolger, Lohse & Usadel, 2014*). Sequences from a subset of samples were assembled with Velvet version 1.2.10 (*Zerbino & Birney, 2008*) to establish a range of suitable k-mer lengths for sequence assembly. We initially attempted optimal assemblies using various criteria with VelvetOptimiser version 2.2.5 (*Gladman & Seemann, 2012*). However, because some regions of the salmonid genome are tetrasomic, and thus do not meet the ploidy assumptions made in VelvetOptimiser, we established a range of k-mer lengths, 57–83 sites representing between 1/2 and 2/3 of total read length, and assembled salmonid sequences from all odd k-mer values within that range. We evaluated each of the resulting assemblies for the presence of UCEs using the PHYLUCE pipeline (*Faircloth, 2015*) and retained only the assembly that contained the maximum number of UCE loci, following the approach described in *Campbell et al. (2017)*.

We identified homologous UCE loci and prepared them for alignment using the PHYLUCE pipeline. The PHYLUCE package screens out reciprocally duplicate enriched loci (i.e., potential paralogs) and performs a multiple sequence alignment of each of the remaining loci using MAFFT version 7.130b (*Katoh et al., 2002*; *Katoh & Standley, 2013*; *Katoh & Toh, 2008*). We screened tetrasomic loci in our dataset by searching previously published assemblies of Atlantic salmon (*Salmo salar*) (*Lien et al., 2016*) and rainbow trout (*Pearse et al., 2019*) for the UCEs sequenced in our study. Within Atlantic salmon, nine pairs of chromosome arms were indicated by *Lien et al. (2016)* that may be tetrasomic, seven of which were identified LORe regions by *Robertson et al. (2017)*. From rainbow trout, seven chromosome pairs that were clearly tetrasomic in *Pearse et al. (2019)* were screened constituting homeologs to the seven LORe pairs screened in Atlantic salmon (*Blumstein et al., 2019*). BLASTN (*Altschul et al., 1990*) alignments of assembled UCEs that presented two alignments of >95% overlap and >97% similarity in the tetrasomic regions of either Atlantic salmon or rainbow trout were considered tetrasomic. UCEs that were not assembled and found to be single copy for all targeted species were also removed to create a "100% coverage alignment."

To understand the effects of relaxing strict filtering of loci on phylogenetic inference, we generated four additional datasets: (1) a "95% coverage alignment" allowing one missing taxon per locus, (2) a "93% coverage alignment" allowing up to two missing taxa per locus, (3) a "75% coverage alignment" allowing up to five missing taxa per locus, (4) a "tetrasomic loci alignment" including only tetrasomically inherited loci permitting 5 missing taxa per locus.

## Phylogenetic inference with the multispecies coalescent

We conducted joint GT–ST estimation of each of the five datasets with the StarBEAST2 Bayesian framework (*Ogilvie, Bouckaert & Drummond, 2017*) as implemented in BEAST version 2.4.8 (*Bouckaert et al., 2014*). For analysis in StarBEAST2, our data sets were partitioned by each UCE locus and the Akaike Information Criterion (AIC) was applied to select the best-fitting model of nucleotide substitution for each locus in each of our datasets using ModelGenerator version 0.85 (*Keane et al., 2006*). We ran several iterations of BEAST with varying length and sampling frequency and evaluated convergence of the runs and effective sample sizes (ESS) with Tracer version 1.6 (*Rambaut et al., 2018*) to determine proper sampling frequency and MCMC chain run length. We required that ESS values be 200 or greater to assure an adequate sample size for our posterior parameter estimations. Analysis of our initial BEAST runs showed that the estimates of parameter values failed to converge for datasets with more than 13 UCEs even after 300 million to 2 billion MCMC generations and showed evidence of over-parameterization, such as a lack of convergence in each chain and low consistency in LnL/prior values. To account for this lack of convergence in the larger datasets and test the effects of using a simpler nucleotide substitution model, we conducted an additional set of phylogenetic inferences in BEAST of all of our datasets and partition schemes thereof but specified the HKY+$\Gamma$ nucleotide substitution model with four categories of gamma distributed rate variation and empirical base frequencies for each partition. For each of our analyses, we combined the independent chains and applied a burnin with LogCombiner version 2.1.3, and generated a maximum clade credibility tree using TreeAnnotator version 2.1.2, specifying a posterior probability (PP) limit of 0.50 and median node heights.

In addition to the joint GT–ST estimation conducted in BEAST, we also estimated the ML tree of each UCE locus and used summary coalescence analyses to estimate a species tree from the resulting set of gene trees a posteriori. We conducted this analysis in addition to the joint GT and ST analysis conducted in StarBEAST2 because this approach can show high accuracy and is computationally more tractable than a full probabilistic approach (*Mallo & Posada, 2016*). For each of our datasets, we estimated the ML tree of each UCE locus using RAxML version 8.0.19, specifying the GTR+$\Gamma$ model of sequence evolution. We passed these trees to ASTRAL version 5.5.9 and MP-EST version 2.2.0, which each estimated a species tree from the collection of UCE loci.

## Concatenated analyses

A concatenated maximum likelihood (ML) analysis was conducted with each data set (100%, 95%, 93%, 75% coverage alignments and the tetrasomic loci alignment).

We conducted both partitioned (i.e., by gene, or objectively defined) and un-partitioned analyses under the GTR+Γ model of sequence evolution using RAxML version 8.2.10 (*Stamatakis, 2014*) with rapid bootstrap stopping. We used the program PartitionFinder version 1.1.1 (*Lanfear et al., 2012*) to generate an objective optimal partitioning strategy with partitions of individual UCE loci specified with the following options: the GTR+Γ model, greedy algorithm (*Lanfear et al., 2014*) and likelihoods estimated by RAxML (–raxml option). If the objective partition strategy differed from either unpartitioned or partitioned strategies, a third analyses of that data set was undertaken with RAxML.

## Salmonid subfamily relationships

Though the composition and monophyly of salmonid subfamilies is well-established (*Norden, 1961*; *Sanford, 1990*), the relationships between them remain contentious. However, if we assume that each subfamily is truly monophyletic, and that their relationships take the form of a bifurcating tree, then there are only three possible arrangements of the salmonid subfamilies. This tractable number of potential combinations allows us to consider each possible arrangement of the taxa explicitly and test for the most preferred arrangement given our data. To avoid test-specific biases, we evaluated the support for each possible relationship of the subfamilies using four different approaches: (1) triplet analysis of independently estimated gene trees, (2), Bayes factors computed from the posterior trees of StarBEAST2 analyses, (3), a test under the multi-species coalescent with the program MP-EST, and, (4), approximately unbiased (AU) tests of concatenated trees.

### Triplet analysis

A triplet analysis is based around a three-taxon rooted tree. In this manuscript, the three salmonid subfamilies form the basis of the test. Expectations with triplet analyses are that given the three possible relationships present with a rooted three-taxon case, the correct relationship should predominate over roughly equal portions of the alternative hypotheses (*Campbell, Chen & López, 2014*; *Cranston, 2010*; *Pamilo & Nei, 1988*). We performed triplet analysis using only the most stringent (i.e., no tetrasomic loci, no missing taxa) dataset, the 100% coverage alignment. For each UCE locus, a salmonine, coregonine, and thymalline were chosen at random with *Esox lucius* as a fixed outgroup. We inferred a ML for each set of taxa with RAxML and the tree stored. We repeated the process of random draws of three taxon sets 1,000 times for each locus and calculated the final frequency of relationships among the three salmonid subfamilies.

### Bayes factors

We compared the strength of evidence for different salmonid subfamily relationships using Bayes factors calculated from the StarBEAST2 post-burnin trees (*Jeffreys, 1935*; *Kass & Raftery, 1995*). We assumed a uniform prior for these calculations, that is, each of the three subfamily relationships were equally likely. A Bayes factor is then the ratio of the posterior probabilities between two alternatives, and here, we present ratios of posterior probabilities of C+T/C+S, C+T/S+T and C+S/S+T.

**Table 1 Characteristics of UCE alignments.** For the five alignments generated for the study, the amount of missing data allowed in terms of taxa/locus is given, the number of resulting UCEs generated, and the number that were considered tetrasomic are presented. The resulting alignment length, unique alignment patterns and percentage of gaps and undetermined portions are then provided.

| Alignment name | Missing data threshold | Number of UCEs | Number of tetrasomic loci | Number of loci analyzed | Alignment length | Unique alignment patterns | Gaps and undetermined (%) |
|---|---|---|---|---|---|---|---|
| 100% Coverage | No missing taxa/locus | 8 | 2 | 6 | 2,331 | 645 | 15.63 |
| 95% Coverage | 1 taxon/locus | 19 | 6 | 13 | 5,297 | 1,261 | 21.18 |
| 93% Coverage | 2 taxa/locus | 42 | 13 | 28 | 10,232 | 2,382 | 21.89 |
| 75% Coverage | 5 taxa/locus | 138 | 33 | 105 | 36,835 | 8,688 | 33.83 |
| Tetrasomic loci | 5 taxa/locus | 138 | 33 | 33 | 12,758 | 2,661 | 33.13 |

### Tests under multispecies coalescent

We used a likelihood ratio test, incorporating the multispecies coalescent with MP-EST version 2.2.0 (*Liu, Yu & Edwards, 2010*) and the R package phybase version 2.0 (*Liu & Yu, 2010*) available at https://github.com/lliu1871/phybase. We supplied unconstrained gene trees from our summary coalescence analyses to MP-EST and constrained the species tree topologies but allowed the branch lengths to be optimized to best fit the data presented by the gene trees. We modified the test2.sptree function of phybase (provided in Data Supplement, https://doi.org/10.25338/B8DC81) to enable all three subfamily arrangements to be compared as described in *Liu et al. (2019)*. Likelihood ratio test statistics (*t*) were calculated between pairs of alternative trees and we generated a distribution of the test statistic by bootstrap sampling gene trees 100 times to estimate the null distribution for *t* and provide a measure of statistical significance (*p*-value).

### Approximately unbiased tests

We conducted topological tests of the three possible relationships of salmonid subfamilies in a concatenated framework by constraining subfamily relationships for different alignments and partitioning schemes and passing per site likelihoods from RAxML version 8.2.11 to the program CONSEL version 0.20. We used CONSEL to compute the AU test from these per site likelihoods (*Shimodaira & Hasegawa, 2001*).

## RESULTS

### Sequence alignment characteristics

Of the 500 UCE loci that we targeted, the number of total captured loci and the number of putatively single copy loci sequenced from each taxon varied (Table S1). For diploid outgroups, *Esox* and *Umbra*, 415 and 409 of the captured UCEs, respectively, were determined to be single-copy. Within salmonids, 166–275 of the captured loci were indicated to be single copy. Overall, we retained only eight single-copy UCE loci across all sampled taxa. Two of these loci were identified as coming from tetrasomically-inherited regions of the salmonid genome, and so were removed. Thus only six out of 500 candidate UCE loci passed the most stringent screening process and were present in all targeted taxa for a total of 2,331 aligned nucleotide sites (Table 1). The datasets that we generated

when allowing for missing data yielded more retained UCE loci, ranging between 13 and 105, with aligned lengths between 5,297 and 36,835 aligned nucleotide sites (Table 1). We identified 33 tetrasomically inherited loci for analysis in the "tetrasomic loci alignment" (see Table 1) and this dataset contains a total of 12,758 aligned nucleotide sites. Across all datasets, the proportion of gaps and other ambiguously alignment regions ranged from 15.63% to 33.83%.

## Model selection and MCMC convergence

For each locus in the "100% coverage" alignment (see Table 1), we initially specified the model of nucleotide substitution that received the lowest AIC score from ModelGenerator (Table S2). We obtained sufficient ESS and convergence of the parameter estimates with three independent MCMC chains of 400 million generations, sampled every 40,000 generations, with a 10% burn-in. For the "95% coverage" dataset, we specified nucleotide substitution models for each locus which received the lowest AIC score in our analysis using ModelGenerator. We obtained sufficient ESS and convergence with two chains of 50 million generations, sampled every 5,000 generations, with a 10% burn-in. For all remaining datasets, analyses with models of nucleotide substitution specified by ModelGenerator failed to generate sufficient ESS and convergence of parameter estimates in our MCMC chains. In response, we broadly reduced parameterization by simplifying nucleotide substitution models.

Analyses of the "93% coverage" alignment, the "75% coverage", and "tetrasomic loci" alignments generated sufficient ESS and convergence of the parameter estimates with two independent MCMC chains of 200 million generations sampled every 20,000 generations, three chains of 800 million generations sampled every 80,000 generations, and four chains of 200 million generations sampled every 20,000 generations, respectively, and in each case we applied a 10% burn-in.

## Phylogenetic inference

The results of joint GT-ST analyses are shown in Figs. 4 and 5 and summarized in Table 2. The clade Coregoninae + Thymallinae (C + T) is found in our 100% coverage alignment with ModelGenerator-specified nucleotide substitution models with strong support (0.98 PP) and in all analyses with reduced coverage datasets with very weak support (0.34–0.52 PP; see Table 2). Analysis of the tetrasomic loci alignment strongly supports Coregoninae + Salmoninae (C + S, 0.99 PP). The monophyly of *Coregonus* is not supported in the results of the 100% coverage species-tree, but is weakly supported (0.43–0.45 PP) in two of three of our reduced coverage analyses and strongly supported in our tetrasomic loci species-tree (1.00 PP). *Salmo* is inferred as the sister group of *Parahucho* in the 100% coverage species-tree with strong support (0.99 PP) and in all the reduced coverage analyses (0.98–0.99 PP). In contrast, the species tree inferred from the tetrasomic loci alignment places *Oncorhynchus* and *Parahucho* as sister lineages (0.38 PP).

The results of the summary coalescent analyses conducted in ASTRAL support a sister group relationship between Salmoninae and Thymallinae (4/5 analyses), but with low

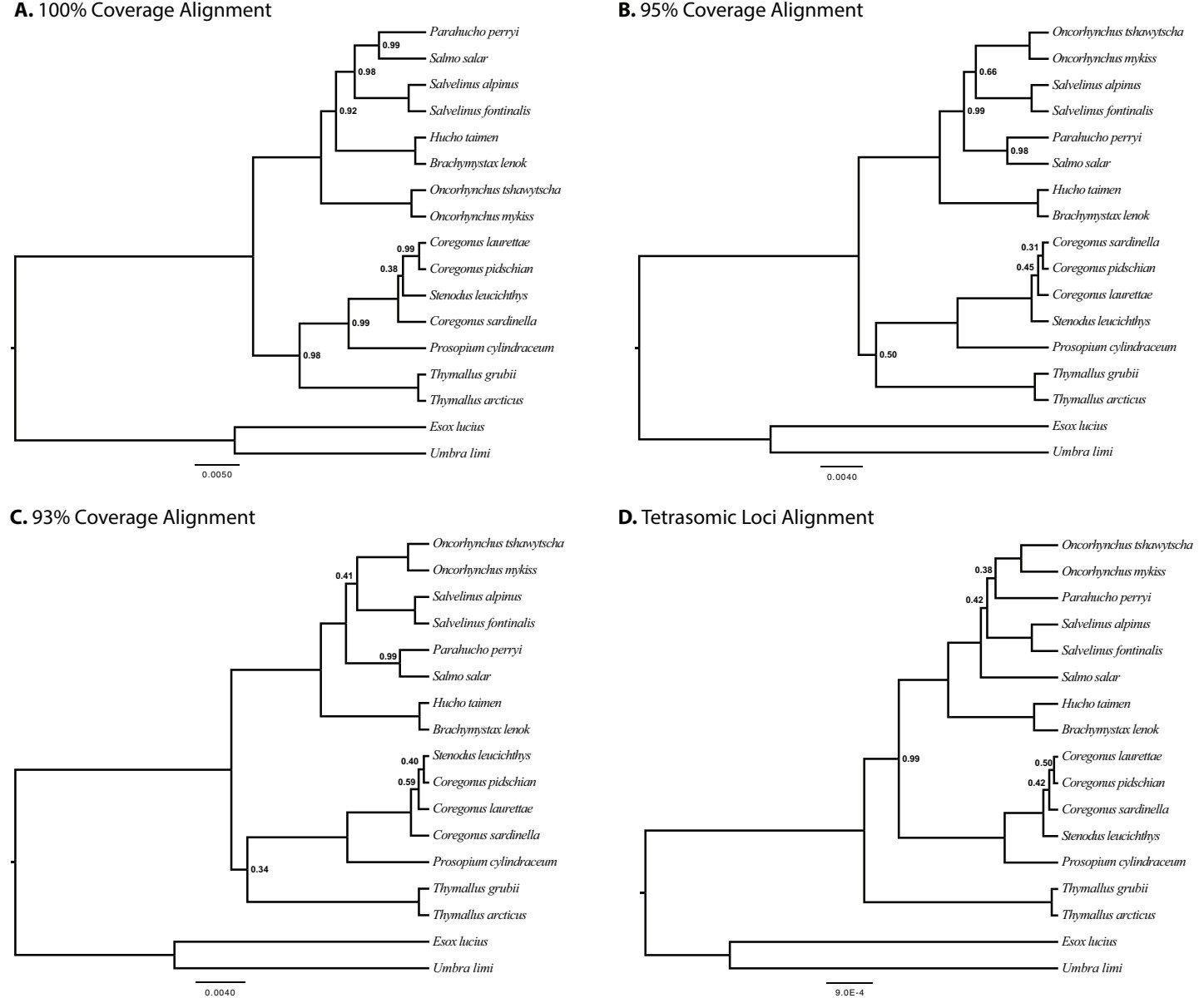

**Figure 4 Species trees generated in StarBEAST2.** A joint estimation of species trees and gene trees is presented with posterior probability presented at nodes for four separate alignments. Posterior probabilities equal to 1.00 are not shown. (A) A total of 100% coverage alignment. (B) A total of 95% coverage alignment. (C) A total of 93% coverage alignment. (D) Tetrasomic loci alignment.

support (PP ranges from 0.38 to 0.57), and, from tetrasomic loci, Coregoninae + Salmoninae (0.91 PP, see Table 2). Monophyly of *Coregonus* is supported in the results of 3/5 of the analyses, though with only weak support (PP ranges from 0.68 to 0.74). Within Salmoninae, *Parahucho* is variously supported as sister to several different taxa with weak support (BPP < 0.70), and to a *Oncorhynchus* + *Salvelinus* clade with moderate support from tetrasomic loci (0.85 BPP, see Table 2). The results of the summary coalescent analyses conducted in MP-EST are inconsistent and reported in Table 2.

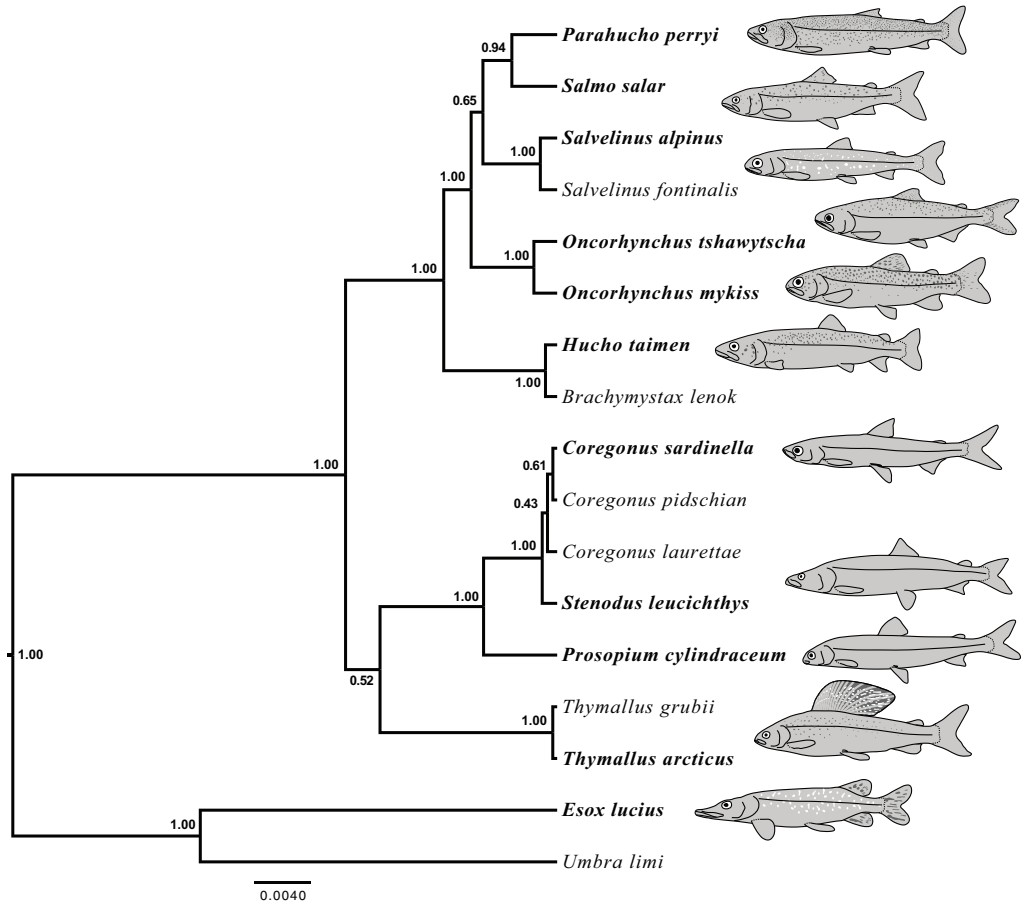

**Figure 5 Species tree generated in StarBEAST2 from 75% coverage alignment.** A joint estimation of species trees and gene trees is presented with posterior probability presented at nodes. Depictions of species in the phylogeny are presented with bold species names indicating the represented species. *Hucho taimen* is represented by a drawing of *Hucho hucho*. Drawings by Thaddaeus Buser.

## Concatenated analyses

Concatenated ML analyses supported either Coregoninae + Salmoninae or Salmoninae + Thymallinae relationships depending on the number of loci and partitioning strategy (Table S3, trees provided in Data Supplement). Partitioning strategy of alignments only altered inferred subfamily relationships in one case (95% coverage alignment), but caused substantial changes in bootstrap support (BS) values. Monophyly of *Coregonus* was only supported in three of thirteen analyses (BS range 70–100%), and the sister of *Parahucho* varied from *Salmo* (5/13 analyses, BS range 24–100%), to *Oncorhynchus* (5/13 analyses, BS range 50–99%) to *Salvelinus* (3/13 analyses, BS range 54–67%).

## Salmonid subfamily relationships

A majority (50.79%) of trees generated in the triplet analyses support a Coregoninae + Thymallinae clade. The Salmoninae + Coregoninae (31.31%) and Salmoninae + Thymallinae (17.90%) clades are found in smaller but substantial proportions of the

**Table 2 Outcomes of phylogenetic analyses incorporating the multispecies coalescent.** For the three methods employing the multispecies coalescent, (A) StarBEAST2, (B) ASTRAL, and (C) MP-EST, summaries of the results are presented for each of the five alignments. The three key relationships examined in the study, subfamily relationships, monophyly of *Coregonus*, and the sister lineage of *Parahucho* are presented along with support values if available.

| Alignment | Number of independently modeled loci | Sister subfamilies | Sister subfamilies posterior probability | Monophyly of *Coregonus* | Support for monophyly of *Coregonus* | Sister of *Parahucho* | Support for sister of *Parahucho* |
|---|---|---|---|---|---|---|---|
| (A) StarBEAST2 analyses | | | | | | | |
| 100% Coverage | 6 | Coregoninae + Thymallinae | 0.98 | No | – | *Salmo* | 0.99 |
| 95% Coverage | 13 | Coregoninae + Thymallinae | 0.50 | Yes | 0.45 | *Salmo* | 0.98 |
| 93% Coverage | 28 | Coregoninae + Thymallinae | 0.34 | No | – | *Salmo* | 0.99 |
| 75% Coverage | 105 | Coregoninae + Thymallinae | 0.52 | Yes | 0.43 | *Salmo* | 0.98 |
| Tetrasomic loci | 33 | Coregoninae + Salmoninae | 0.99 | Yes | 1.00 | *Oncorhynchus* | 0.38 |
| (B) ASTRAL analyses | | | | | | | |
| 100% Coverage | 6 | Salmoninae + Thymallinae | 0.42 | Yes | 0.74 | (*Salmo* + (*Oncorhynchus* + *Salvelinus*)) | 0.69 |
| 95% Coverage | 13 | Salmoninae + Thymallinae | 0.38 | Yes | 0.68 | (*Salmo* + (*Oncorhynchus* + *Salvelinus*)) | 0.68 |
| 93% Coverage | 28 | Salmoninae + Thymallinae | 0.57 | No | – | *Salmo* | 0.46 |
| 75% Coverage | 105 | Salmoninae + Thymallinae | 0.50 | No | – | (*Oncorhynchus* + *Salvelinus*) | 0.54 |
| Tetrasomic loci | 33 | Coregoninae + Salmoninae | 0.91 | Yes | 0.53 | (*Oncorhynchus* + *Salvelinus*) | 0.85 |
| (C) MP-EST analyses | | | | | | | |
| 100% Coverage | 6 | Coregoninae + Salmoninae | | Yes | | *Oncorhynchus* | |
| 95% Coverage | 13 | Salmoninae + Thymallinae | | Yes | | (*Salmo* + *Oncorhynchus*) | |
| 93% Coverage | 28 | Salmoninae + Thymallinae | | No | | *Salmo* | |
| 75% Coverage | 105 | Coregoninae + Thymallinae | | No | | *Oncorhynchus* | |
| Tetrasomic loci | 33 | Coregoninae + Salmoninae | | No | | (*Oncorhynchus* + *Salvelinus*) | |

remaining trees. Bayes factors from the 100% coverage alignment indicate that a sister group relationship of Coregoninae and Thymallinae should be strongly preferred. Bayes factors for all other datasets when comparing C + T to alternatives do not provide strong

**Table 3 Hypothesis testing with the multispecies coalescent.** The testing of subfamily relationships either through (A) Bayes factor (StarBEAST2), or, (B) the *t*-statistic (MP-EST, described in text) are shown. Partitioning and modeling of each of the five alignments is summarized along with the number of loci analyzed. Significance values are presented along with the *t*-statistic and are in bold face if significant.

| (A) | StarBEAST2 Analyses | Partitioning and modeling scheme | Number of independently modeled loci | Bayes factor C+T vs C+S | Bayes factor C+T vs S+T | Bayes factor C+S vs S+T |
|---|---|---|---|---|---|---|
| | 100% Coverage alignment | By UCE, ModelGenerator | 6 | 131.47 | 108.84 | 0.82 |
| | 95% Coverage alignment | By UCE, ModelGenerator | 13 | 1.82 | 2.21 | 1.21 |
| | 93% Coverage alignment | By UCE, HKY+Γ | 28 | 1.00 | 1.01 | 1.00 |
| | 75% Coverage alignment | By UCE, HKY+Γ | 105 | 1.23 | 9.33 | 7.57 |
| | Tetrasomic loci alignment | By UCE, HKY+Γ | 33 | 0.00 | 0.81 | 248.20 |

| (B) | MP-EST analyses | Partitioning and modeling scheme | Number of independently modeled loci | *t*-statistic CT vs CS | *p*-value | *t*-statistic CT vs ST | *p*-value | *t*-statistic CS vs ST | *p*-value |
|---|---|---|---|---|---|---|---|---|---|
| | 100% Coverage alignment | By UCE | 6 | 14.80 | 0.27 | 11.69 | 0.37 | 3.10 | 0.28 |
| | 95% Coverage alignment | By UCE | 13 | $5.60 \times 10^{-5}$ | 0.49 | 0.61 | 0.47 | 0.61 | 0.41 |
| | 93% Coverage alignment | By UCE | 28 | 4.74 | 0.39 | 2.26 | 0.27 | 7.01 | 0.27 |
| | 75% Coverage alignment | By UCE | 105 | 11.25 | 0.28 | 10.58 | 0.18 | 0.67 | 0.36 |
| | Tetrasomic loci alignment | By UCE | 33 | 130.18 | 0.06 | $8.10 \times 10^{-4}$ | 0.45 | 130.18 | **0.05** |

support for C + T (Table 3). MP-EST analysis indicated that a sister group relationship for C + S was significantly supported via likelihood ratio tests for the tetratomic loci; no other data sets could distinguish among alternative topologies. The concatenated AU test results were not significant with all 100% and 95% coverage alignment tests, indicating that the alternative C + T hypothesis could not be rejected in analyses of the reduced data sets (Table S3).

# DISCUSSION

## Salmonid subfamily relationships

A sister group relationship of Coregoninae and Thymallinae (C + T) is broadly preferred to alternative topologies in our analyses (e.g., Fig. 5). This relationship is recovered in the joint GT–ST estimations of all data sets that exclude tetrasomically-inherited loci (PP = 0.34–0.98) and is also the preferred hypothesis indicated by Bayes factors and the triplet analysis. While no concatenated results support C + T, this result is obtained with as few as six UCE loci in joint GT-ST estimation with declining support as the number of loci increased. A Coregoninae + Thymallinae clade has been identified in previous molecular studies that either implicitly or explicitly account for potential polyploidy in their data. The C + T clade was recovered by analyses of mitochondrial genomes, which are not affected by issues of paralogy and can be expected to be less prone to ILS due to their haploid mode of transmission (*Campbell et al., 2013*; *Horreo, 2017*). Likewise, constructing phylogenies with both copies of Ss4R duplicated genes, an explicit approach, leads to support of a sister Coregoninae + Thymallinae (*Macqueen & Johnston, 2014*; *Robertson et al., 2017*).

Our results do not support the conclusions of previous studies that examined morphological characters of the salmonids and unanimously placed Thymallinae as the sister group to Salmoninae (*Kendall & Behnke, 1984*; *Sanford, 1990*; *Stearley & Smith, 1993*; *Wilson & Li, 1999*). Furthermore, our results combined with mitogenomic and other studies explicitly addressing polyploidy strongly support C + T over the two alternative hypotheses apparent in molecular phylogenetic studies of salmonids (*Alexandrou et al., 2013*; *Crête-Lafrenière, Weir & Bernatchez, 2012*; *Shedko, Miroshnichenko & Nemkova, 2012*). In light of growing molecular evidence conflicting with the morphology-based hypothesis, a critical reassessment of the morphological evidence that supports that relationship is needed but is beyond the scope of the present study.

Though C + T is the preferred grouping in our most stringent analyses, not all of the inference strategies implemented here supported this relationship. Analysis of the concatenated dataset and GT-ST estimation using ASTRAL produced inconsistent results and generally did not support the C + T clade. The inconsistent results between summary coalescence approach and joint estimation of the species tree and gene trees may indicate loss of a weak phylogenetic signal in the summarization step. Alternatively, the phylogenetic signal may have been blurred as a result of treating all UCE loci as a single partition evolving under a common substitution mode. The triplet analysis did not meet our prediction that, given three possible topologies, the true relationship should predominate and the alternative, incorrect arrangements should receive approximately equal portions of support. The triplet analyses yielded the putatively incorrect topologies at unequal frequencies (31.30% and 17.90%). This could be a result of the relatively small size of our "100% coverage" alignment or be a product of resampling selecting certain individuals frequently due to few representatives in some subfamilies (two thymallinine species are in the dataset).

## Monophyly of *Coregonus*

Our results inconsistently support the monophyly of *Coregonus* (e.g., Figs. 4 and 5). Though the joint GT–ST analysis of tetrasomic loci and the ML analysis of the concatenated 100% coverage data set shows strong support for a monophyletic *Coregonus*, all other analyses that recovered the clade did so with weak support (PP < 0.75, BS = 71%, see Table 2). Even within a given analytical framework, support for monophyly of this genus varied. For example, MP-EST analyses showed a monophyletic *Coregonus* for the two most stringent datasets, but not in the analysis of any others. The uncertainty in this relationship within our data set may be derived from the relatively young age (i.e., ~10 mya, *Horreo, 2017*) of the clade that includes *Stenodus* and *Coregonus* resulting in low variation in the conserved UCE loci that were indentified to be single copy across a broad range of salmonid species. Additionally, during the analytical study design of this manuscript, a chromosomal-level assembly of a coregonine was unavailable leading to uncertainty in the tetrasomic status of loci in this subfamily based of off salmonine genome assemblies.

Previous studies examining the monophyly of *Coregonus* with respect to *Stenodus* have yielded conflicting results (*Bernatchez, Colombani & Dodson, 1991*; *Bodaly et al., 1991*;

*Crête-Lafrenière, Weir & Bernatchez, 2012*; *Horreo, 2017*; *Sajdak & Phillips, 1997*; *Vuorinen et al., 1998*). Most recently, analysis of mitogenomic data has shown *Coregonus* to be monophyletic, with *Stenodus* as its sister taxon (*Horreo, 2017*). However, the phylogenetic placement of the enigmatic *Coregonus huntsmani*, a divergent member of the genus, remains untested in any mitogenomic or phylogenomic study.

*Crête-Lafrenière, Weir & Bernatchez (2012)* found *C. huntsmani* to be the sister lineage to *Stenodus* and all other *Coregonus* species and earlier isozyme and mitochondrial restriction data show *C. huntsmani* to be distinct from the other main *Coregonus* subgenera (*Coregonus* and *Leucichthys*) but did not include *Stenodus* in the sampling (*Bernatchez et al., 1991*). We examined publicly available COI data from coregonines including two haplotypes from *C. huntsmani* and found that species to be sister group to a monophyletic *Coregonus* with low support (BS = 58%, alignment tree and methods are supplied in the Data Supplement). At present molecular data from *C. huntsmani* are very limited, however including this taxon in future datasets will be critical for conclusively testing the monophyly of *Coregonus* and its relationship with *Stenodus*. Further investigations of phylogenetic relationships of *Coregonus* and *Stenodus* should also consider the role and prevalence of hybridization within *Coregonus* and between *Stenodus* and *Coregonus*, (*McPhail, 2007*) as the signatures of hybridization and ILS are difficult to disentangle and our analyses assumed such signatures derived from ILS (*Yu et al., 2011*).

## Placement of *Parahucho*

The Sakhalin taimen *Parahucho perryi* is a rare species and the sole member of its genus (*Rand, 2006*; *Vladykov & Gruchy, 1972*). This critically endangered fish occupies a limited geographic range and has a narrow range of suitable habitats in Northern Japan and the Russian Far East (*Fukushima et al., 2011*; *Kimura, 1966*; *Rand, 2006*). The placement of *Parahucho* as sister to *Salmo* is strongly supported by all joint GT–ST analyses of non-tetrasomic loci (4 of 4, 0.98–0.99 PP) and by concatenated analysis of our 100% and 95% coverage alignment (BS = 66–100%). The original description of Sakhalin taimen placed it as a congener of Atlantic salmon (*Salmo salar*), as *Salmo perryi* (*Brevoort, 1856*). Subsequently, the similarities of Sakhalin taimen to the European huchen *Hucho hucho* were noted, and these two species were placed in the same genus, *Hucho* (*Günther, 1868*; *Jordan & Snyder, 1902*). *Hucho* and *Parahucho* are both large bodied piscivores, but differ in that the latter has 10 less vertebrae, basibranchial teeth, a median set of teeth on the supralingual, and 70–100 less scales in the lateral line (*Vladykov, 1963*; *Vladykov & Gruchy, 1972*). Indeed, the two genera are not closely allied in any of the results presented here, nor has a close relationship of the two genera been reported in any other molecular phylogenetic studies (*Campbell et al., 2013*; *Crespi & Fulton, 2004*; *Matveev, Nishihara & Okada, 2007*; *Oakley & Phillips, 1999*). Though there has been considerable disagreement in the placement of *Parahucho*, the emerging consensus from nuclear gene sequences supports a *Parahucho* + *Salmo* sister relationship (*Lecaudey et al., 2018*), as is recovered here.

### Phylogenetic results from disomic and tetrasomic loci

Resolving subfamily relationships was a primary aim of this study, but our results repeatedly showed a conflict in this segment of the phylogeny between analyses of the most stringently assembled dataset (i.e., 100% coverage) and of the remaining alignments, and particularly of the tetrasomic loci. For example, in GT–ST analyses conducted with StarBEAST2 of the 100%, 95%, 93% and 75% coverage alignments, Coregoninae and Thymallinae are supported as sister lineages. Analyses of the 100% coverage data set yield very strong (0.98 PP) support for that grouping, which declines substantially in analyses of the other datasets (0.34–0.52 PP). On the other hand, the 33 tetrasomic loci we removed strongly supported a Coregoninae + Salmoninae clade (0.99 PP). Likewise, we observed strong support for a sister relationship between *Salmo* and *Parahucho* in the more reduced datasets (0.98–1.00 PP), and less support in the more extensive 75% alignment (0.94 PP). The topology of the species tree inferred from StarBEAST2 analysis of tetrasomic loci indicates that *Salmo* and *Parahucho* are not closely related (Fig. 4D). In summary, two of three sets of alternative placements we considered in detail (i.e., subfamily relationships, *Coregonus* monophyly, and placement of *Parahucho*) in this study yield conflicting results when comparing datasets of loci with differing ploidy levels. We also find that support values for the C + T clade decline with greater proportions of missing data in the underlying dataset.

We attempted to identify tetrasomic loci in the sequenced dataset through the assembly of a single contig matching a UCE locus that was placed within genomic locations of known tetrasomy in rainbow trout and Atlantic salmon. While there is evidence for broad conservation of homologous tetrasomic genomic regions in salmonids (*Brieuc et al., 2014*; *Kodama et al., 2014*; *Lien et al., 2011*; *Sutherland et al., 2016*), there is also evidence of substantial lineage-specific rediploidization in salmonids and polyploids in general taxa (*Robertson et al., 2017*; *Spoelhof, Soltis & Soltis, 2017*). Our definition of tetrasomic regions from two salmonines is taxonomically restrictive and may have allowed the inclusion of regions of tetrasomic inheritance in other species sequenced. Consequently, increasing the proportion of missing data may have introduced paralogous loci into our analyses due to lineage-specific rediploidization processes (e.g., Fig. 3B).

## CONCLUSION

In salmonids, ancestral autopolyploidy and the resulting residual tetrasomy increases the potential for conflicts between gene trees and species trees due to a greater likelihood for incomplete lineage sorting. Lineage-specific rediploidization processes also are known to be concentrated in the same tetrasomically pairing chromosome regions contributing to the analysis of paralogous loci. Applying stringent criteria aimed at minimizing these problems yielded a highly reduced dataset comprising six UCE loci. These loci show clear support for a Coregoninae + Thymallinae clade and for the placement of *Parahucho* sister to *Salmo*, but do not support the monophyly of *Coregonus* in joint GT and ST analysis. Expanding the number of aligned loci by lowering stringency of filtering resulted in generally reduced confidence in clade makeup and support for conflicting phylogenetic

relationships while increasing support for the monophyly of *Coregonus*. The use of tetrasomic loci likewise resulted in an alternative topology from that of our 100% coverage analysis, and contradictory relationships may exhibit high support values. The accurate inference of salmonid phylogenies is challenged not only by limits of data and the sophistication of analyses, but also in part due to the evolutionary history and patterns of the lineage.

## ACKNOWLEDGEMENTS

We would like to thank Scott Edwards and Lang Liu for providing code and a manuscript in advance of publication along with helpful advice for hypothesis testing under the multispecies coalescent. We would also like to thank Max Belasco for his assistance generating the sequence data used in this study. Michio Fukushima kindly allowed us to use a picture of *Parahucho perryi*, Armando Piccinini and Stefano Porcellotti who allowed us to use a picture of *Hucho hucho*, and Beatrice Smith who allowed us to use a picture of *Stenodus leucichthys* to create a depiction of each of these species for our Fig. 5. We would like to recognize Kevin McCracken (University of Miami) for his support in promoting Next-Generation Sequencing and Bioinformatics and the University of Alaska Fairbanks from which this data set was generated. Tissues in this study were provided by the University of Alaska Museum of the North, the Burke Museum (University of Washington) and the Academy of Natural Sciences of Drexel University.

### Funding

This study was funded by a Genomics Seed Grant from the Alaska Idea Network for Biomedical Research (INBRE) awarded to Matthew A. Campbell and J. Andrés López. Research reported in this publication was supported by an Institutional Development Award (IDeA) from the National Institute of General Medical Sciences of the National Institutes of Health under grant number P20GM103395. The funders had no role in study design, data collection and analysis, decision to publish, or preparation of the manuscript.

### Grant Disclosures

The following grant information was disclosed by the authors:
Genomics Seed Grant from the Alaska Idea Network for Biomedical Research (INBRE).
Institutional Development Award (IDeA) from the National Institute of General Medical Sciences of the National Institutes of Health: P20GM103395.

### Competing Interests

The authors declare that they have no competing interests.

### Author Contributions

- Matthew A. Campbell conceived and designed the experiments, performed the experiments, analyzed the data, prepared figures and/or tables, authored or reviewed drafts of the paper, and approved the final draft.
- Thaddaeus J. Buser analyzed the data, prepared figures and/or tables, authored or reviewed drafts of the paper, and approved the final draft.
- Michael E. Alfaro conceived and designed the experiments, authored or reviewed drafts of the paper, and approved the final draft.
- J. Andrés López conceived and designed the experiments, analyzed the data, prepared figures and/or tables, authored or reviewed drafts of the paper, and approved the final draft.

## DNA Deposition

The following information was supplied regarding the deposition of DNA sequences:

The sequence data are available at GenBank under BioProject PRJNA603774 as a Targeted Locus Study: KDUN00000000, KDUO00000000, KDUP00000000, KDUQ00000000, KDUR00000000, KDUS00000000, KDUT00000000, KDUU00000000, KDUV00000000, KDUW00000000, KDUX00000000, KDUY00000000, KDUZ00000000, KDVA00000000, KDVB00000000.

## Data Availability

Assembled sequence data matching UCE loci are available both from NCBI: PRJNA603774 and at Dryad: Campbell, Matthew; Buser, Thaddaeus; Alfaro, Michael; Lopez, J. Andres (2020), Data from: Addressing incomplete lineage sorting and paralogy in the inference of uncertain salmonid phylogenetic relationships, v3, UC Davis, Dataset, DOI 10.25338/B8DC81.

Alignments, tree files, scripts used in the triplet analysis and hypothesis testing under the multispecies coalescent are available in the Supplemental Files.

Accession numbers and originating tissue collection are available in Table S4. Specimens are housed at ANSP, FMNH, UAMN or UWFC.

Phybase is available at GitHub: https://github.com/lliu1871/phybase. The altered function is available as a Supplemental File.

## Supplemental Information

Supplemental information for this article can be found online at http://dx.doi.org/10.7717/peerj.9389#supplemental-information.

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
