# Peer review of "Addressing incomplete lineage sorting and paralogy in the inference of uncertain salmonid phylogenetic relationships"

_PeerJ, doi:10.7717/peerj.9389_

## Round 0.1 · original submission · Minor Revisions

The article has been deemed of high quality by both reviewers. However they both request minor changes and clarifications in the text. Please try to take into account their suggestions.

Reviewer 1 has worthwhile minor suggestions and underlines a need for clarification on the partitioning scheme.

Reviewer 2 has more comments, both clarification and replacing their work on UCE in a larger context of usefulness for other studies and taxa, and some background additions on UCEs, duplicated regions and their challenges. Indeed, as s/he suggests, the strongest points of this paper are less on the phylogeny of the group itself and more on the theoretical and analytical subtelties resulting from incomplete lineage sorting and genome duplication, and therefore are more widely applicable to other taxa, and this could be better highlighted. For the phylogenetic results, slightly more complete comparisons to results of other datasets would indeed be valuable. Last, s/he suggests valuable development and clarifications in various places in the manuscript.

Reviewer 1 ·

Basic reporting

This manuscript was well written and provided proficient review of relevant literature to support the claims. It has informative figures and appropriate associated supplemental data. Additionally it includes tables that help to follow the results of the abundant analyses conducted.

The authors make clear the purpose of their study, the history of the problems associated with Salmonidae and relevant taxon information, and provide an concise conclusion to their study supported by the data and results of there numerous analyses.

I only have very minor suggestions for Lines 228-229: I would simply add "under the assumption the subfamilies are truly monophyletic" in addition something along the lines of "which appears to be a sound assumption (citations)".

Experimental design

The authors have done an excellent job of providing rigorous phylogenetic analyses including coalescent-based methodology as well can concatenation-based. They have also experimented with data-matrix completely and tested different ways of partitioning the data used in analysis. Furthermore they have conducted phylogenetic topology testing. The level of phylogenetic, data set, and topology support testing is exceptional.

The only minor comment I have on methodology is on line 187: it is unclear how partitioning was accomplished for StarBEAST2 multispecies coalescent analyses. It is stated that substitution models were assigned using ModelGenerator, but was each locus treated as single partition? Or the entire alignment treated as a single partition.

This is certainly not required give the current rigor of the methods, but the authors may consider making use of the new UCE specific partitioning scheme, the Sliding-Window Site Characteristics, developed by Tagliacolo & Lanfear. Here is the reference: Tagliacollo, V.A. & Lanfear, R. (2018) Estimating improved partitioning schemes for ultraconserved elements. Molecular Biology and Evolution, 35, 1798–1811.

Validity of the findings

Given the extensive testing conducting, the authors conclusions are well-founded. The conclusions are concisely stated and are all well-supported by the analyses conducted.

Additional comments

The methodological rigor of this manuscript is above and beyond what I have seen exhibited in many recent phylogenomic studies. Additionally, the manuscript was well-written and includes highly informative figures that help make clear the necessity for such rigor. It was a pleasure reviewing your manuscript and your work is highly commendable.

Reviewer 2 ·

Basic reporting

Addressing incomplete lineage sorting….salmonids

The ms is applies a set of UCE loci to selection of salmonid species with the aim of resolving several higher order phylogenetic questions. Much attention is given to identifying problems related to incomplete lineage sorting, paralogy and tetrasomic inheritance. The manuscript is very well written, and analysis carried out at a very professional level. The conclusions are mostly sound and well formulated, but their relevance to the body of literature as well as future studies is not clarified.
Additionally, the importance of both the questions and the conclusions, considering that similar or identical results have been already obtained, when sufficiently large data sets have been used is marginal. In reality, the conclusions that the authors reached are based on a very small set (N = 6) of loci due to their stringent filtering and this raises the question of whether the applied loci set are more broadly applicable outside of this data set? Additionally, the overall structural style and emphases outlined in the introduction could be improved. The authors could perhaps balance the marginal importance of the data or conclusions by moving the analytical aspects of the ms more into the foreground, and address the potential applications of their analysis for other studies or study groups (see also comments below). I also have a few minor questions concerning methodology, which I pose below.

Introduction
The ms poses at first very large questions (concerning e.g. the tree of life), and then discusses at length teleost duplication events, and above all gene-tree – species tree conflicts. However, the study is rather focused on high-level relationships (sub-families or genera) of salmonid fishes, examined with a somewhat unique (UCE) set of loci. In my view the abstract and introduction should more quickly prepare the reader for the focus of the article, and spend some time arguing precisely why this set of loci, and the accompanying analytical approaches may or may not be appropriate for the questions and problems at hand. For example, the authors do not spend any time presenting the advantages or disadvantages of UCE loci? Some background on why these markers were chosen and what their characteristics are, is required. What potential does this set of loci have for addressing similar questions in other groups?

At the end of the introduction the authors pose three phylogenetic questions concerning salmonid fishes, the relationships among sub-families, the monophyly of Coregonus, and the placement of Parahucho. However, a major emphasis of the analysis, relates to theoretical and/or analytical hurdles concerning incomplete lineage sorting and genome duplication, and not the problem of resolving these relationships, which have been addressed in numerous publications. Additionally, a major dilemma with the conflicting reports concerning these relationships relates to marker choice, which is slowly being overcome with NGS based data sets. The authors should more specifically address this. Another problem is taxon sampling, which the authors ignore somewhat (except for discussing one specific example, C. huntsman). I do not get the feeling at the end of the ms, that the reader is well informed as to why these conflicts in results have occurred, and to what extent has this data set resolved the issue, as opposed to larger genome-wide sampling? For example, has the application of mtDNA-based phylogenies for these questions led to misleading or congruent results? Has the application of nuclear markers to these questions led to misleading or congruent results? Does (or would) the application of a large number (1000s) of nuclear markers overcome the problems that the authors outline, or is it necessary to consider incomplete lineage sorting and duplicated regions in all attempts at phylogenetic analysis of salmonid fishes, regardless of the number of markers applied? How does use of UCEs help in resolving these problems? The authors also mention “lineage-specific rediploidization” at the last sentence of the introduction (and again in the conclusions) with no explanation of what that is and its potential influence on phylogenetic analysis. It is doubtful that most readers understand this concept, and whether or not it is being specifically addressed in the manuscript? If this is one of the major hurdles posed for correct phylogenetic inference in salmonids (as the authors state), it would seem very important to know if they have successfully overcome this problem?

Methods
Authors correctly identify the problem with tetrasomic loci in salmonids, and attempt to address this problem by identifying a sub-set of their UCE loci in Atlantic salmon and rainbow trout that presumably lie in duplicated regions. Are these regions specifically the so-called LORe regions discussed in Robertson et al 2017? Please specify. For the uninformed reader, it would appear that this problem only relates to duplication, but it is more complicated than that. The authors should explain the lineage-specific nature of these events and how it relates to the WGD, and specifically how this then causes a problem for phylogenetic inference. Lastly, the question of whether or not the identification of these regions in Atlantic salmon is sufficient for identifying such loci throughout the family is saved for comments in the discussion, but the problem is known beforehand and not a result of this analysis. Thus the authors should comment on this in the methods.


Conclusions
Here the authors mention “lineage-specific rediploidization” as a potential problem in their own data set (or at least with the number of loci increased beyond 6?). This relates then to my comment on the sentence at the end of the discussion – elsewhere in the ms, this term does not appear.
The authors state that tetrasomic regions exist on 8 pairs of chromosome arms in salmonids. Can they really say this? Has this been explicitly shown outside of Atlantic salmon? Perhaps for rainbow trout as well, but in other salmonids? Considering the wide range in karyotype between both general and species, should this statement not be more narrowly defined to those groups where these regions have been specifically identified?

Experimental design

all comments contained under basic reporting.

Validity of the findings

all comments contained under basic reporting.

Additional comments

all comments contained under basic reporting.

---

## Round 0.2 · accepted · Accept

The work on the manuscript took into account all the remarks of the two reviewers. It has clarified some points and made for a very good manuscript.